# Preliminary Results of ERAS Protocol in a Single Surgeon Prospective Case Series

**DOI:** 10.3390/medicina58091234

**Published:** 2022-09-06

**Authors:** Gabriele Tulone, Nicola Pavan, Alberto Abrate, Ettore Dalmasso, Piero Mannone, Davide Baiamonte, Sofia Giannone, Rosa Giaimo, Marco Vella, Carlo Pavone, Riccardo Bartoletti, Vincenzo Ficarra, Alchiede Simonato

**Affiliations:** 1Section of Urology, Department of Surgical Oncological and Stomatological Sciences, University of Palermo, 90127 Palermo, Italy; 2Urology Unit, Department of Surgery, ASST Valtellina e Alto Lario, 23100 Sondrio, Italy; 3Urology Unit, Department of Surgery, S. Croce e Carle Hospital, 12100 Cuneo, Italy; 4Department of Translational Research and New Technologies in Medicine and Surgery, University of Pisa, 56124 Pisa, Italy; 5Urology Unit, Department of Surgery, University of Messina, 98125 Messina, Italy

**Keywords:** enhanced recovery after surgery, outcomes, ERAS, radical cystectomy, bladder cancer

## Abstract

*Background and Objectives*: The aim was to compare the intra and postoperative outcomes between the Enhanced Recovery After Surgery (ERAS) protocol versus the standard of care protocol (SCP) in patients who underwent radical cystectomy performed by a single surgeon. *Materials and Methods:* A retrospective comparative study was conducted including patients who underwent radical cystectomy from 2017 to 2020. Length of stay (LOS), incidence of ileus, early postoperative complications, and number of re-hospitalizations within 30 days were considered as primary comparative outcomes of the study. *Results:* Data were collected for 91 patients who underwent cystectomy, and 70 and 21 patients followed the SCP and ERAS protocol, respectively. The mean age of the patients was 70.6 (SD 9.5) years. Although there was a statistically significant difference in time to flatus (TTF) [3 (2.7–3) vs. 1 (1–2 IQR) days, *p* < 0.001, in the SC hospital and in the ERAS center respectively], no difference was reported in time to first defecation (TTD) [5 (4–6) vs. 4 (3–5.8), *p* = 0.086 respectively]. The median LOS in the SCP group was 12 (IQR 11–13) days vs. 9 (IQR 8–13 *p* = 0.024). In the postoperative period, patients reported 22 complications (37% in SCP and 42.8% in ERAS group, *p* = 0.48). *Conclusions:* The study reveals how even partial adherence to the ERAS protocols leads to similar outcomes when compared to SCP. As a single surgeon series, our study confirmed the role of surgeons in reducing complications and improving surgical outcomes.

## 1. Introduction

Radical cystectomy (RC) with pelvic lymph node dissection is the gold standard treatment for muscle-invasive bladder cancer [1,2,3]. This procedure, despite the surgical experience gained over the years, is not free from complications both during the operative practice and peri- and postoperative time. The interest of research in the urological field is focused on finding a common guideline for the management of patients who undergo RC to improve the postoperative recovery. Enhanced recovery after surgery (ERAS) [4,5], theorized by Kehlet, consists of evidence-based interventions applied during pre-, intra- and postoperative time to reduce the physical stress due to surgery, to accelerate patient rehabilitation, and reduce hospitalization time. Although the ERAS protocol was born for colorectal surgery, several studies have shown a possible role of this protocol, even in the case of RC, the postoperative recovery [6]. This study aims to compare patients who underwent cystectomy who followed the ERAS protocol with patients who followed the usual care protocol analyzing the return to normal bowel functions, length of stay, and early- and late-postoperative complication incidence.

## 2. Materials and Methods

A retrospective comparative study was conducted including consecutive patients who underwent radical cystectomy in two urological centers. We collected the data regarding the hospitalization of patients undergoing RC, with an open approach in a standard care center from 2017 to 2019 (non-ERAS protocol), and the data of patients from a second center undergoing the same surgery, but with a certified ERAS protocol. The ERAS certification was achieved in 2020 (ERAS items are present in Table 1, according to the ERAS guidelines). Inclusion criteria were consecutive patients who underwent RC for muscle invasive bladder cancer performed by the same surgeon. For each patient, all of the preoperative characteristics such as age, sex, comorbidities, blood tests, ECOG (Eastern Cooperative Oncology Group), ASA (American Society of Anesthesiologists), and CCI (Charlson Comorbity Index) were considered. We compared the postoperative parameters of the two centers (ERAS and non-ERAS) analyzing the differences in length of stay (LOS), time to flatus (TTF), time to defecation (TTD), presence of ileus, positioning of the nasogastric tube (NGT) and its maintaining time, onset of complications, early or late, using the Clavien–Dindo (CD) score, and the presence of re-hospitalizations. The study was conducted on the available retrospective data and according to our ethical committee, approval is not required.

### Statistical Analysis

The means and standard errors of the mean were reported for continuous variables assumed to be normally distributed. The remaining continuous variables were summarized by their median values and interquartile ranges. Data are compared with the Student’s *t*-test, Mann–Whitney, and Chi-square for the normal, non-normal, and nominal variables, respectively. A value of *p* < 0.05 was considered significant. The sample size was not calculated because of the retrospective nature of the study.

## 3. Results

Data were collected from 91 consecutive patients who underwent cystectomy, 70 of them followed the standard care protocol (SCP) and 21 the ERAS protocol. The mean age of the patients was 70.6 ± 9.5 years. Pre-operative blood analyses of the two subgroups did not demonstrate statistically significant differences except for the albumin serum levels, which was higher in the SPC center (Table 2. There was a statistically significant difference between the incidences of comorbidities, with a greater rate of in-patients treated with the SCP (78.5% vs. 28.5%, *p* < 0.001). The performance status was evaluated with the ECOG score (Table 3. The ASA score was higher in the SCP patients (ASA ≥ 3: 97.1% vs. 47.6%). Operating time (OR) and estimated blood loss (EBL) were also reported in Table 2. Although there was a statistically significant difference in TTF (3 (2.7–3) vs. 1 (1–2) *p* < 0.001, in standard care hospital vs. ERAS hospital), no difference were recorded in the two groups for the time to first defecation (TTD) (5 (4–6) vs. 4 (3–5.8), *p* = 0.086, respectively) (Figure 1). The percentage of patients without postoperative NGT is significantly different between the two groups (10.2% vs. 90.4% in non-ERAS centers and in the ERAS patients, respectively). This can be explained by the fact that in the SPC center, the NGT was routinely inserted and maintained after the surgical procedure. There was a difference in the ileus rate, with a higher incidence in the ERAS center (23.8 % vs. 4.1 %, *p* = 0.037) (Table 4). The difference can be explained by the fact that NGT was routinely inserted and maintained in the SPC center, reducing the rate of postoperative nausea. The median length of stay (LOS) of patients in the non-ERAS center was 12 days (IQR 11–13) and in the ERAS center, it was 9 days (IQR 8–13.4) (*p* = 0.024). In the postoperative period, 22 (37%) complications have been recorded in the non-ERAS center (68% grade 1 and 2 according to the Clavien–Dindo classification), while nine (43%) complications were recorded in the ERAS center (100% grade 1, 2) without significant differences between the two groups (*p* = 0.48). Table 4 reports the rate of postoperative complications according to CD classification. According to these findings, the incidence rate of late postoperative complications was significantly higher in the ERAS patients compared to the usual care patients (35% vs. 12.2%, respectively; *p* = 0.03). The mean follow-up time after surgery was 15.8 (IQR 7.3–21.8) months and 7 (IQR 4.3–9.8) in the SCP and ERAS groups, respectively (*p* = 0.01).

## 4. Discussion

In all surgical areas, there is increasing attention on the hospitalization time, influenced by the onset of postoperative complications. The ERAS protocol was born to improve the clinical outcomes of surgery and reduce the economic pressure of patients in many developing countries [7,8]. Since the introduction of ERAS into colorectal surgery, several preoperative, intraoperative, and postoperative steps have been implemented to improve patient rehabilitation. The benefits of the ERAS protocols are similar in many surgeries such as bariatric, esophageal, and other gastrointestinal surgeries [9,10,11]. In urological practice, the use of the ERAS protocol is limited for one of the most challenging interventions, both for surgeons and patients, RC. [4,12]. The ERAS protocol consists of 21 items including pre-, intra-, and postoperative approach [6]. The literature refers to many patients undergoing radical cystectomy that benefit from enhancements in perioperative management. Unfortunately, sometimes, the protocol is hard to apply for its rigidity. The aim of our study was to evaluate the difference in outcome in the usual care patients comparing the ERAS patients and the effectiveness of the ERAS protocol by comparing specific parameters. Our data indicate that there was no substantial improvement in the postoperative course between patients who followed the ERAS protocol and those undergoing usual care. The population examined in the non-ERAS center was characterized by higher comorbidities and higher ASA score. Although the TTF was shorter in patients undergoing the ERAS protocol [3 days (2.7–3) vs. 1 day (1–2) *p* < 0.001, respectively], the same cannot be said for the TTD time, which was comparable. The ERAS protocols were also introduced to reduce postoperative ileus [13]. There are many definitions of the ileus in the literature. We defined post operative ileus after radical cystectomy, the reduction in bowel movement, intolerance of oral diet, and vomiting, requiring cessation of oral intake with the necessity of nasogastric tube insertion [14]. There is no univocal criterion for the insertion of NGT, and it is always upon clinical judgment. Standardization would be helpful to compare the results of future studies on postoperative ileus [15]. Our data demonstrated a lower incidence of the ileus in the usual care population compared to the ERAS protocol population. Despite this result, the TTF was significantly shorter in the ERAS group. Another purpose of the ERAS protocol is to minimize the hospitalization time: such a target was confirmed in our study. Furthermore, the hospitalization time is influenced by postoperative complications evaluated with the CD classification, both in the short- and long-term. Furthermore, for this parameter, there is no evidence of substantial and statistically significant differences. Although the ERAS protocol is today of great interest in urological surgery, however, there are conflicting data in the literature on its ability to influence perioperative outcomes [15,16]. Many studies have shown that ERAS courses can reduce the length of stay [15,17,18,19,20,21], while some refute this thesis [21,22,23,24]; the same was true in the evaluation of the reduction in time to the recovery of bowel activity and in the readmission rate [19,20,21,22,23,25,26]. The study does not want to discredit the ERAS protocol, but to rather understand whether there are unassessed parameters that could influence the postoperative course. These parameters can be subjective and hardly standardizable as, for instance, the surgeon experience and the number of interventions by the center. The surgical technique, the manual skills, and the number of interventions can influence the operative course as they affect various parameters such as the operative time, the time of ileal–ileal anastomosis and the suture technique, blood losses, and the incidence of intraoperative complications. The high number of interventions of a center is associated with a greater confidence with the type of patient, with their needs, knowledge of time for the postoperative steps such as mobilization, which is essential for a rapid recovery of the patient, and prompt identification of deviations from the patient in a normal postoperative course. This hypothesis was already explored by previous studies and in 2016, Moschini et al. reported the published results into a systematic review [27,28]. According to their results, the surgical volume is affecting outcomes after radical cystectomy in terms of perioperative complications. The main limitation of this study was the initial retrospective design. A larger number of patients needs to be included in future trials to reach significance and stronger results. However, we reported this number of patients because this is exactly the required amount necessary to obtain the ERAS certification.

## 5. Conclusions

The application of the ERAS protocol in patients undergoing RC should improve the timing of rehabilitation and reduce LOS and the hospitalization costs. Our study surprisingly showed that our surgical technique resized the results expected from the ERAS Society protocol to accelerate the postoperative recovery time and reduce the early- and late-postoperative complications. Therefore, a precise, accurate, and fast surgical procedure with reduced blood loss that tends to minimize postoperative complications is able to determine the patient’s recovery on its own, regardless of strict protocols. In other words, the ERAS protocols are certainly useful to speed up recovery, but does not seem to be able to disregard a good surgical technique.

## Figures and Tables

**Figure 1 medicina-58-01234-f001:**
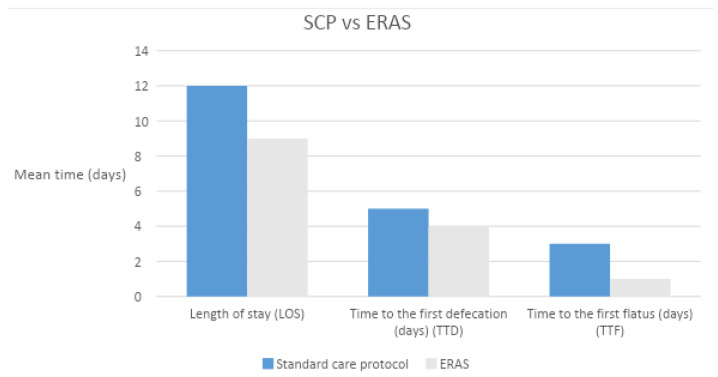
The standard care protocol vs. the ERAS protocol: LOS, TTD, TTF.

**Table 1 medicina-58-01234-t001:** A summary of each ERAS item and their respective level of evidence and grade of recommendation derived from both the cystectomy and the colorectal literature.

ERAS Single Item	Summary	Specifics for Cystectomy Patients/Open Questions	Evidence for Cystectomy/Rectal Surgery	Grade of Recommendation
1. Preoperative counseling and education	Patients should receive routine dedicated preoperative counseling and education	Surgical details, hospital stay, and discharge criteria in oral and written form; stoma education; patient’s expectations.	Na/Low	Strong
2. Preoperative medical optimization	Preoperative optimization of medical conditions should be recommended. Preoperative nutritional support should be considered, especially for malnourished patients	Correction of anemia and co-morbidities; Nutritional support; Smoking cessation and reduction of alcohol intake 4 weeks prior to surgery; encouraging physical exercise.	Na/Moderate Na/High Na/Moderate Na/Very low	
3. Oral mechanical bowel preparation	Preoperative bowel preparation can be safely omitted	/	Moderate/High	Strong
4. Preoperative carbohydrates loading	Preoperative oral carbohydrate loading should be administered to all non-diabetic patients	/	Na/Low	Strong
5. Preoperative fasting	Intake of clear fluids up until 2 h before induction of general anesthesia is recommended. Solids are allowed up until 6 h before anesthesia.	/	Na/Moderate	Strong
6. Preanasthesia medication	Avoidance of long-acting sedatives	/	Na/Moderate	Strong
7. Thrombosis prophylaxis	Patients should wear well-fitting compression stockings, and receive pharmacological prophylaxis with LMWH. Extended prophylaxis for 4 weeks should be carried out in patients at risk. 12 h interval between injections and epidural manipulation	Cystectomy patients are considered at risk; prolonged prophylaxis should therefore be administered	Na/High	Strong
8. Epidural analgesia	Thoracic epidural analgesia is superior to systemic opioids in relieving pain. It should be continued for 72 h	/	Na/High	Strong
9. Minimally invasive approach	At most feasible; in trial setting Long term oncological results awaited	Laparoscopic/robotic cystectomy is not recommended outside a trial setting until long term results are available.	Low/Moderate	Strong
10. Resection site drainage	Perianastomotic and/or pelvic drain can be safely omitted.	Because of urine leak, drainage might be required in cystectomy patients.	Na/Low	Weak
11. Antimicrobial prophylaxis and skin preparation	Patient should receive a single dose antimicrobial prophylaxis 1 h before skin incision. Skin preparation with chlorexidine-alcohol prevents/decreases surgical site infection.	/	Na/High Na/Moderate	Strong
12. Standard anesthetic protocol	To attenuate the surgical stress response, intraoperative maintenance of adequate hemodynamic control, central and peripheral oxygenation, muscle relaxation, depth of anesthesia, and appropriate analgesia is recommended. Fast acting agents?	/	Na/Moderate	Strong
13. Perioperative fluid management	Fluid balance should be optimized by targeting cardiac output using the esophageal Doppler system or other systems for this purpose and avoiding overhydration. Judicious use of vasopressors is recommended with arterial hypotension.	High-risk patients need close and individualized goal directed fluid management. There are several ways to achieve this and all must be used together with sound clinical judgment.	Low/High	Strong
14. Preventing intraoperative hypothermia	Normal body temperature should be maintained per-and postoperatively.	Especially relevant for cystectomy patients since operative duration is prolonged	Na/high	Strong
15. Nasogastric intubation	Postoperative nasogastric intubation should not be used routinely.	Early removal is recommended.	Low/High	Strong
16. Urinary drainage	Transurethral catheter can be remove on postoperative day 1 after pelvic surgery in patients with a low risk of urinary retention.	Ureteral stents and transurethral neo-bladder catheter should be used. The optimal duration of ureteral stenting (at least until POD 5) and transurethral catheterization is unknown.	Very low/Low	Weak
17. Prevention of postoperative ileus	A multimodal approach to optimize gut function should involve gum chewing and oral magnesium.	/	Moderate/Moderate	Strong
18. Prevention of PONV	A multimodal PONV prophylaxis should be adopted in all patients with ≥2 risk factors.	Multimodal prophylaxis.	Very low/Low (High in high-risk patients)	Strong
19. Poperative analgesia	A multimodal postoperative analgesia	/	Very low/Low (High in high-risk patients)	Strong
20. Early mobilization	Early mobilization should be encouraged	2 h out of bed POD 0; 6 h out of bed POD 1.	Na/High	Strong
21. Early oral diet	Early oral nutrition should be started 4 h after surgery	/	Na/Moderate	Strong
22. Audit	All patients should be audited for protocol compliance and outcome	Routine audit of outcomes, cost-effectiveness, compliance and changes in protocol.	Na/Low	Strong

Na: not available; ERAS: enhanced recovery after surgery; LMWH: low molecular weight heparin; PONV: postoperative nausea and vomiting; POD: postoperative day.

**Table 2 medicina-58-01234-t002:** The patient characteristics.

Variable	Total	Standard	ERAS	*p*-Value
Patients	91	70 (76.9)	21 (23.1)	
Age, year	70.6 ± 9.5	69.3 ± 9.6	74.9 ± 7.7	0.016
BMI, kg/m^2^	26.2 ± 4.2	26.4 ± 4.3	25.7 ± 4.1	0.538
Gender				
Female	13 (14.3)	10 (14.2)	3 (14.3)	0.722
Male	78 (85.7)	60 (85.8)	18 (85.7)	
ECOG performance status				
0	27 (33.7)	23 (32.8)	4 (40)	0.801
1	51 (63.7)	45 (64.2)	6 (60)	
2	2 (2.5)	2 (3)	0 (0)	
Comorbidities				
No	30 (33.0)	15 (21.4)	15 (71.4)	<0.001
Yes	61 (67.0)	55 (78.6)	6 (28.6)	
Hypertension				
No	47 (51.6)	30 (42.8)	17 (81)	0.005
Yes	44 (48.4)	40 (57.2)	4 (19)	
Diabetes				
No	63 (69.2)	46 (65.7)	17 (81)	0.290
Yes	28 (30.8)	24 (34.3)	4 (19)	
COPD				
No	76 (83.5)	55 (78.5)	21 (100)	0.047
Yes	15 (16.5)	15 (21.5)	0 ()	
Cardiopathy				
No	66 (72.5)	46 (65.7)	20 (95.2)	0.017
Yes	25 (27.5)	24 (34.3)	1 (4.8)	
ASA score				
2	13 (14.3)	2 (2.8)	11 (52.3)	<0.001
3	67 (73.6)	57 (81.4)	10 (47.7)	
4	11 (12.1)	11 (15.8)	0 (0)	
Charlson CI	6 (5–6)	6 (5–6)	6 (4.6–6.4)	0.913
OR, min	273 (212–334)	286 (223–348)	231 (194–268)	<0.001
EBL, mL	399 (380–588)	431 (239–623)	290 (157–423)	<0.001

Continuous variables were expressed as the mean ± SD or median (IQR); nominal variables were expressed as No. (%). ERAS: enhanced recovery after surgery; BMI: body mass index; ECOG: Eastern Cooperative Oncology Group performance status; COPD: Chronic obstructive pulmonary disease; ASA: American Society of Anesthesiologists; OR: operating time EBL: estimated blood loss.

**Table 3 medicina-58-01234-t003:** The pre-operative blood analyses.

Variable	Total	Standard	ERAS	*p*-Value
Hb, g/dL	13.0 ± 1.8	13.1 ± 1.8	12.8 ± 1.7	0.596
Neutrophils, ×10^3^/µL	5.6 ± 2.5	5.8 ± 2.5	4.9 ± 1.9	0.287
Lymphocytes, ×10^3^/µL	1.9 ± 0.6	1.9 ± 0.6	1.9 ± 0.6	0.920
Platelets, ×10^3^/µL	271 ± 98	271 ± 100	269 ± 86	0.946
Creatinine, mg/dL	1.1 ± 0.8	1.2 ± 0.9	1.0 ± 0.3	0.103
Albumin, g/L	39.0 ± 6.2	39.9 ± 6.0	32.5 ± 3.2	<0.001

Continuous variables were expressed as the mean ± SD or median (IQR); nominal variables were expressed as No. (%).

**Table 4 medicina-58-01234-t004:** The postoperative characteristics.

Variable	Total	Standard	ERAS	*p*-Value
Length of stay (LOS), days	11 (10–13)	12 (11–13)	9 (8–13.4)	0.024
Time to the first defecation (TTD), days	5 (4–5.3)	5 (4–6)	4 (3–5.8)	0.086
Time to the first flatus (TTF), days	2 (2–3)	3 (2.7–3)	1 (1–2)	<0.001
Ileus				
No	63 (90.0)	47 (95.9)	16 (76.2)	0.037
Yes	7 (10.0)	2 (4.1)	5 (23.8)	
NGT ( nasogastric tube) placement				
No	21 (28.0)	7 (12.5)	14 (73.6)	<0.001
<48 h	25 (33.3)	21 (37.5)	4 (21)	
>48 h	29 (38.7)	28 (50)	1 (5.4)	
Postoperative NGT				
No	24 (32.0)	5 (9.2)	19 (90.4)	<0.001
Yes	51 (68.0)	49 (90.8)	2 (9.6)	
Early complications				
No	60 (65.9)	48 (68.5)	12 (57.1)	0.480
Yes	31 (34.1)	22 (31.5)	9 (42.9)	
Early complications Clavien–Dindo				
No	58 (63.7)	47 (67.1)	11 (52.3)	0.098
1	7 (7.7)	6 (8.5)	1 (4.7)	
2	17 (18.7)	9 (11.1)	8 (38.3)	
3	6 (6.6)	6 (8.5)	0 ()	
4	1 (1.1)	1 (1.4)	0 ()	
5	2 (2.2)	1 (1.4)	1 (4.7)	
Early complications time, days	4 (2.2–5.4)	3 (0–8)	5 (4–6)	0.376
Late complications				
No	57 (82.6)	43 (87.7)	14 (70)	0.157
Yes	12 (17.4)	6 (12.3)	6 (30)	
Late complications Clavien–Dindo				
No	56 (81.2)	43 (87.7)	13 (65)	0.003
1	1 (1.4)	0 ()	1 (5)	
2	3 (4.3)	0 ()	3 (15)	
3	7 (10.1)	6 (12.3)	1 (5)	
4	0	0 ()	0 ()	
5	2 (2.9)	0 ()	2 (10)	
Late complications time, days	38.5 (29.7–122.6)	34.5 (22.3–157.1)	51	0.505

Continuous variables were expressed as the mean ± SD or median (IQR); nominal variables were expressed as No. (%).

## Data Availability

The data presented in this manuscript are available on request from the corresponding author.

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
