# Peer review of "Preliminary Results of ERAS Protocol in a Single Surgeon Prospective Case Series"

_medicina, 2022, doi:10.3390/medicina58091234_

Round 1

Reviewer 1 Report

The authors compared the perioperative outcomes of patients with or without ERAS protocol who underwent radical cystectomy. Some comments are described below:

1. From the whole manuscript, It is hard to understand the ERAS protocol. No detail for ERAS protocol contained in this article.

2. Although patients in two different groups were treated by the same surgeon. However, they were from different center. And not the same time. As we know, the learning curve of surgeon would be a factor for surgery outcome.

3. The information of surgery such as operation time, choice of surgical method, transfusion, status of lymph node dissection is also important factor for evaluate the surgery outcome.

4. The number of ERAS group was too small compare to standard group.

Author Response

  1. From the whole manuscript, It is hard to understand the ERAS protocol. No detail for ERAS protocol contained in this article.

Thank you for the observation. We inserted the ERAS items in a table

Table 1

Summary of each ERAS item and their respective level of evidence and grade of recommendation derived from both the cystectomy and the colorectal literature.

ERAS single item

Summary

Specifics for cystectomy patients/ open questions

Evidence for cystectomy/ rectal surgery

Grade of recommendation

1. Preoperative counseling and education

Patients should receive routine dedicated preoperative counseling and education

Surgical details, hospital stay and discharge criteria in oral and written form; stoma education; patient’s

Na/Low

Strong

2. Preoperative medical

Preoperative optimization of medical

expectations

Correction of anemia and

Na/Moderate

Strong

optimization

conditions should be recommended.

Preoperative nutritional support should

be considered, especially for malnourished patients

co-morbidities Nutritional support

Smoking cessation and reduction of alcohol intake 4 weeks prior to surgery;

encouraging physical exercise

Na/High Na/Moderate Na/Very low

3. Oral mechanical bowel preparation

4. Preoperative carbohydrates loading

5. Preoperative fasting

Preoperative bowel preparation can be safely omitted

Preoperative oral carbohydrate loading should be administered to all non-diabetic patients

Intake of clear fluids up until 2 h before

/

/

/

Moderate/High Na/Low

Na/Moderate

Strong Strong

Strong

induction of general anesthesia is

recommended. Solids are allowed up

6. Preanasthesia medication

until 6 h before anesthesia.

Avoidance of long-acting sedatives

/

Na/Moderate

Strong

7. Thrombosis prophylaxis

Patients should wear well-fitting compression stockings, and receive pharmacological prophylaxis with LMWH.

Extended prophylaxis for 4 weeks should

be carried out in patients at risk. 12 h interval between injections and epidural manipulation.

Cystectomy patients are considered at risk; prolonged prophylaxis should therefore be administered

Na/High

Strong

8. Epidural analgesia

9. Minimally invasive approach

Thoracic epidural analgesia is superior to systemic opioids in relieving pain. It should be continued for 72 h

At most feasible; in trial setting

/

Laparoscopic/robotic

Na/High

Low/Moderate

Strong

Strong

Long term oncological results awaited

cystectomy is not recommended outside a trial setting until long

term results are available.

10. Resection site drainage

11. Antimicrobial prophylaxis

Perianastomotic and/or pelvic drain can be safely omitted

Patient should receive a single dose

Because of urine leak, drainage might be required in cystectomy patients

/

Na/Low

Na/High

Weak

Strong

and skin preparation

12. Standard anesthetic protocol

antimicrobial prophylaxis 1 h before skin incision. Skin preparation with chlorexidine-alcohol prevents/decreases surgical site infection.

To attenuate the surgical stress response,

/

/

Na/Moderate

Na/Moderate

Strong

13. Perioperative fluid

intraoperative maintenance of adequate hemodynamic control, central and peripheral oxygenation, muscle relaxation, depth of anesthesia, and appropriate analgesia is recommended. Fast acting agents?

Fluid balance should be optimized by targeting

High-risk patients need close

Low/High

Strong

management

cardiac output using the esophageal Doppler

system or other systems for this purpose

and individualized goal

directed fluid management.

14. Preventing intraoperative

and avoiding overhydration. Judicious use of vasopressors is recommended with arterial hypotension.

Normal body temperature should be

There are several ways to achieve this and all must be used together with sound clinical judgment

Especially relevant for

Na/high

Strong

hypothermia

maintained per-and postoperatively.

cystectomy patients since

operative duration is prolonged

15. Nasogastric intubation

Postoperative nasogastric intubation

Early removal is recommended

Low/High

Strong

16. Urinary drainage

should not be used routinely

Transurethral catheter can be removed

Ureteral stents and transurethral

Very low/Low

Weak

on postoperative day 1 after pelvic surgery in patients with a low risk of urinary retention

neo-bladder catheter should be used. The optimal duration of ureteral stenting (at least until POD 5) and transurethral

catheterization is unknown.

17. Prevention of postoperative ileus

18. Prevention of PONV

19. Poperative analgesia

20. Early mobilization

21. Early oral diet

22. Audit

A multimodal approach to optimize gut function should involve gum chewing and oral magnesium

    A multimodal PONV prophylaxis     should be adopted in all patients with ≥2 risk factors.

A multimodal postoperative analgesia

Early mobilization should be encouraged

Early oral nutrition should be started 4 h after surgery

All patients should be audited for protocol compliance and outcome

/

Multimodal prophylaxis

/

2 h out of bed POD 0

 6 h out of bed POD 1

/

Routine audit of outcomes, cost-effectiveness, compliance and changes in protocol

Moderate/Moderate

Very low/Low (High in high-risk patients)

Very low/Low (High in high-risk patients)

Na/High

Na/Moderate

Na/Low

Strong

Strong

Strong

Strong

Strong

Strong

  1. Although patients in two different groups were treated by the same surgeon. However, they were from different center. And not the same time. As we know, the learning curve of surgeon would be a factor for surgery outcome.

Thanks for observation. The operations were performed by the same surgeon with extensive case history and experience and over a period of four years.

  1. The information of surgery such as operation time, choice of surgical method, transfusion, status of lymph node dissection is also important factor for evaluate the surgery outcome.

Thank for the observation, all these information are reported in table 2, 3 and 4.

  1. The number of ERAS group was too small compare to standard group.

We know that the number of patients undergoing radical cystectomy and who followed the ERAS protocol is small and it is a limitation of our study. However, we reported this number of patients because this is exactly the required amount necessary to obtain the ERAS certification. We stressed this concept also in the discussion.

Reviewer 2 Report

This is a retrospective study of 91 patients who underwent radical cystectomy by a single surgeon. The outcomes were compared between patients who received ERAS (n=21) vs. standard of care (n=70). Despite the difference in time to flatus, no difference was reported in time to the first defecation between the two groups. The median LOS was shorter in the ERAS group (9 vs 12). The 30-day postoperative complications were similar between the two groups. The authors concluded that “the application of ERAS protocol in patients undergoing RC should improve the timing of rehabilitation and reduce LOS and hospitalization costs”.

Herein are my questions and comments:

1.     The sample size is small. It compromises the study power and makes it difficult to draw a meaningful conclusion. In addition, the two groups are different in terms of baseline features and care settings.

2.     The authors should include the details of the ERAS protocol as well as surgical technique. The surgical approach (open vs MIS) and urinary diversion types are not clear. These are all factors that significantly affect surgical outcomes.  

3.     The study lacks multivariable analysis. Given the difference in the baseline features of the patients, it is necessary to do MVA, controlling for confounding factors.

4.     Decreasing LOS to 4-6 days is one of the main outcomes of ERAS. The authors should explain the reason for the long LOS in their study (i.e., 9 days in ERAS and 12 days in non-ERAS).

5.     The conclusion (i.e., improving timing of rehabilitation and reducing hospitalization costs) cannot be drawn from the current findings.

Author Response

This is a retrospective study of 91 patients who underwent radical cystectomy by a single surgeon. The outcomes were compared between patients who received ERAS (n=21) vs. standard of care (n=70). Despite the difference in time to flatus, no difference was reported in time to the first defecation between the two groups. The median LOS was shorter in the ERAS group (9 vs 12). The 30-day postoperative complications were similar between the two groups. The authors concluded that “the application of ERAS protocol in patients undergoing RC should improve the timing of rehabilitation and reduce LOS and hospitalization costs”.

Herein are my questions and comments:

  1. The sample size is small. It compromises the study power and makes it difficult to draw a meaningful conclusion. In addition, the two groups are different in terms of baseline features and care settings.

Thank you for the comment. We know that the number of patients undergoing radical cystectomy and who followed the ERAS protocol is small and it is a limitation of our study but it is also true that to acquire ERAS certification the number of patients needed is equal to that evaluated in the study. The two groups of patients examined are quite homogeneous. They differ in comorbidities, higher in the standard care group and in the ASA score, also higher in the first group. Despite this, the outcomes between the two groups of patients undergoing surgery by the same surgeon are comparable.

  1. The authors should include the details of the ERAS protocol as well as surgical technique. The surgical approach (open vs MIS) and urinary diversion types are not clear. These are all factors that significantly affect surgical outcomes.  

Thank you for the comment. We inserted a table of the ERAS protocol. All patients underwent to radical cystectomy with open approach. We wrote this important information in the manuscript. We didn’t speak about the urinary diversion only because in ERAS protocol the type is not significative.

  1. The study lacks multivariable analysis. Given the difference in the baseline features of the patients, it is necessary to do MVA, controlling for confounding factors.

A multivariate analysis would certainly be useful but it is not possible with the numbers present in the study. For a future study with a larger number of patients it will certainly be performed. This is well reported into the limitation of our preliminary report.

  1. Decreasing LOS to 4-6 days is one of the main outcomes of ERAS. The authors should explain the reason for the long LOS in their study (i.e., 9 days in ERAS and 12 days in non-ERAS).

According to the national health system, in Italy the LOS is longer when compare to USA or UK. This can be addressed by the fact that in Italy patient need to be discharge only when he is autonomous and also able to manage the stoma. This is because in Italy we don’t have a diffuse home-care nursing assistance.

  1. The conclusion (i.e., improving timing of rehabilitation and reducing hospitalization costs) cannot be drawn from the current findings.

We agree with your sentence. Perhaps, we indirectly know that a longer hospital stay and more home care assistance are important factors for the healthcare costs.

Reviewer 3 Report

English language and style are fine/minor spell check required

Author Response

 Thanks for your comment.

Round 2

Reviewer 1 Report

The comments from my past review have been addressed adequately and completely. 

Reviewer 2 Report

Thanks much for revising the manuscript. Unfortunately, the manuscript still has serious issues in terms of sample size and analysis. Additional experiments are needed to improve the quality of this report.